# ENSEMBLE OF LOW-RANK ADAPTERS FOR LARGE LANGUAGE MODEL FINE-TUNING

## ABSTRACT

Fine-tuned LLMs often exhibit poor uncertainty quantification, manifesting as overconfidence, poor calibration, and unreliable prediction results on test data or out-of-distribution samples. One approach commonly used in vision for alleviating this issue is a deep ensemble, which constructs an ensemble by training the same model multiple times using different random initializations. However, there is a huge challenge to ensembling LLMs: the most effective LLMs are very, very large. Keeping a single LLM in memory is already challenging enough: keeping an ensemble of e.g. 5 LLMs in memory is impossible in many settings. To address this issue, we propose an ensemble approach using Low-Rank Adapters (LoRA), a parameter-efficient fine-tuning technique. Critically, these low-rank adapters require a very small number of parameters, orders of magnitude less than the underlying pre-trained model. Thus, it is possible to construct large ensembles of LoRA adapters with almost the same computational overhead as using the original model. We find that LoRA ensembles, applied on its own or on top of pre-existing regularization techniques, gives consistent improvements in predictive accuracy and uncertainty quantification.

## 1 INTRODUCTION

LLMs have demonstrated state-of-art performance in many natural language processing tasks (Radford et al., 2019; Touvron et al., 2023; Brown et al., 2020; Chung et al., 2022; Kojima et al., 2022; OpenAI, 2023). With additional fine-tuning a pre-trained LLM can be adapted to downstream applications or data. However, fine-tuned LLMs can overfit to training data and often exhibit *overconfidence* (as visualized in Fig.1a). Specifically, these models may yield overly certain predictions, especially on incorrectly predicted samples or those from different domains. Ideally, a model should exhibit low confidence when its predictions are likely to be incorrect; otherwise, the outcomes could be dangerously misleading in safety-critical contexts such as medical diagnosis(Singhal et al., 2023), finance (Yang et al., 2023), or decision-making processes (Li et al., 2022).

A widely adopted approach for mitigating overconfidence in deep learning is to make predictions using an *ensemble* of neural networks rather than a single model. There are many approaches for constructing an ensemble of networks, such as training multiple networks with different random initializations (Lakshminarayanan et al., 2017), different hyperparameters (Wenzel et al., 2020b). However, there are two barriers to applying these approaches for fine-tuning LLMs. First, ensembles require storing multiple copies of the model weights and loading them into GPU at test time. This is not practical for modern LLMs. A single LLaMA-13b (Touvron et al., 2023) stored at 16-bit precision, is 25 GB on disk, and loading it to the GPU takes around 6 seconds. In addition, random initialization has been noted to play a crucial role in deep ensembles (Lakshminarayanan et al., 2017; Fort et al., 2019). However, starting the fine-tuning of the individual LLMs with the same initialization – the pre-trained weights – eliminates an important source of randomness and may cause a lack of diversity across the ensemble, thereby potentially reducing its benefits.

Work by Gleave & Irving (2022) and Sun et al. (2022) has attempted building ensembles of fine-tuned LLMs but due to the limitations above, their method is restricted to smaller models such as GPT-2 (Radford et al., 2019) with only 1.5 billion parameters. In this paper, we build on recent advances in efficient LLM fine-tuning with low-rank adapters (LoRA) (Hu et al., 2021) and propose an ensemble method for LLM fine-tuning that scales to models with 13 billion parameters and beyond. We propose

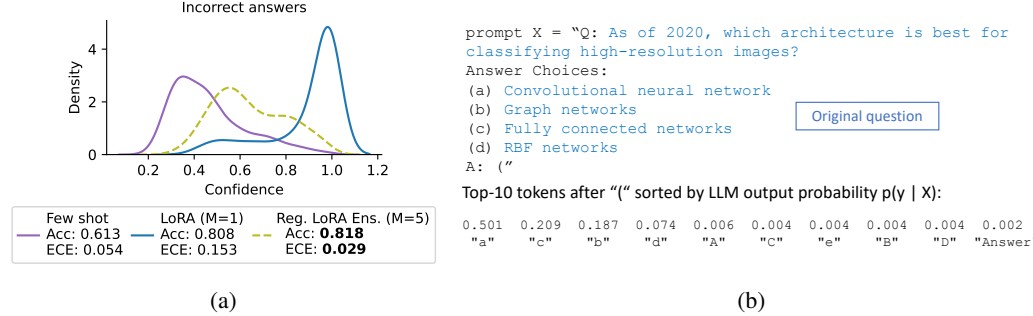

(a)                  (b)

Figure 1: **LoRA ensembles with strong weight decay regularization, is more accurate and better calibrated than fine-tuning a single LoRA component on multiple-choice QA problems such as in Fig. 1b.** Fig 1a, shows a KDE of the confidence with which a pre-trained LLaMA-13b in the few-shot setting (purple line), a fine-tuned LoRA model (blue line), and our proposed LoRA ensembles (yellow dashed line) make wrong predictions on the cqa dataset. The few-shot approach is well-calibrated but often wrong, while LoRA (M=1) is more accurate but overconfident in its wrong predictions. Our approach provides improvements in both accuracy and calibration in terms of ECE.

LoRA ensembles. LoRA ensembles solves the two aforementioned issues: LoRA requires orders of magnitude less storage than the original model: a low-rank adapter for LLaMA-13b is only 30Mb on disk and takes 0.1 seconds to load onto GPU. In addition, the random initialization of the adapter provides the necessary randomness for variability across the ensemble components.

Our empirical results on several commonsense reasoning tasks show that LoRA ensembles improve accuracy and calibration over naive LoRA fine-tuning and produces better ensembles than alternatives based on last-layer fine-tuning (Du et al., 2021) or Monte Carlo dropout (Gal & Ghahramani, 2016).

As an additional contribution we study regularized LoRA ensembles. Classical theory (Breiman, 2001) suggests that the generalization performance of ensembling depends on the diversity of individual components. While no comparable results exist for neural networks it is believed that this intuition still holds for deep ensembles (Lakshminarayanan et al., 2017; Fort et al., 2019). Initialization of the LoRA ensemble components around the same pre-trained weights already introduces a strong correlation between the ensemble components and regularization can further strengthen this effect. Yet we find in an extensive empirical study of LoRA ensembles in combination with different regularization strategies that LoRA ensembles are compatible with regularization and their combination typically further improves prediction and calibration accuracy.

## 2   RELATED WORK

**Robust fine-tuning of language models.** A body of work has proposed regularization methods to improve generalization and calibration during fine-tuning of language models. For instance, He et al. (2022) explores a mix of KL and L2 regularization on the extracted features to retain the calibration of pre-trained masked language models (MLM). Park & Caragea (2022) introduces mixup (Zhang et al., 2017) into MLM fine-tuning, showcasing enhanced calibration at test time. Our approach complements these methods: we can ensemble fine-tuning with any of these techniques (if compatible with LoRA adapters), and we will discuss such strategies extensively in later sections.

**Ensembling of Neural Networks.** Deep ensembles enhance the robustness and reliability of deep learning models (Ovadia et al., 2019). Typically, they are constructed by training the same model with varied initializations (Lee et al., 2015; Lakshminarayanan et al., 2017) or hyper-parameter settings (Wenzel et al., 2020b; Zaidi et al., 2020). Some methods use checkpoints from along the optimization trajectory (Huang et al., 2017) or the Bayesian posterior (Neal, 2012; Zhang et al., 2020; Wenzel et al., 2020a; Izmailov et al., 2021). However, naively applying these methods in the LLM setting requires us to store and load complete model checkpoints which is impractical in many settings due to the very large memory requirements for storing multiple copies of an LLM.

**Ensembling in LLMs.** Two recent papers study ensembling for LLM fine-tuning (Gleave & Irving, 2022; Sun et al., 2022). However, these papers only consider full fine-tuning, optimizing all the

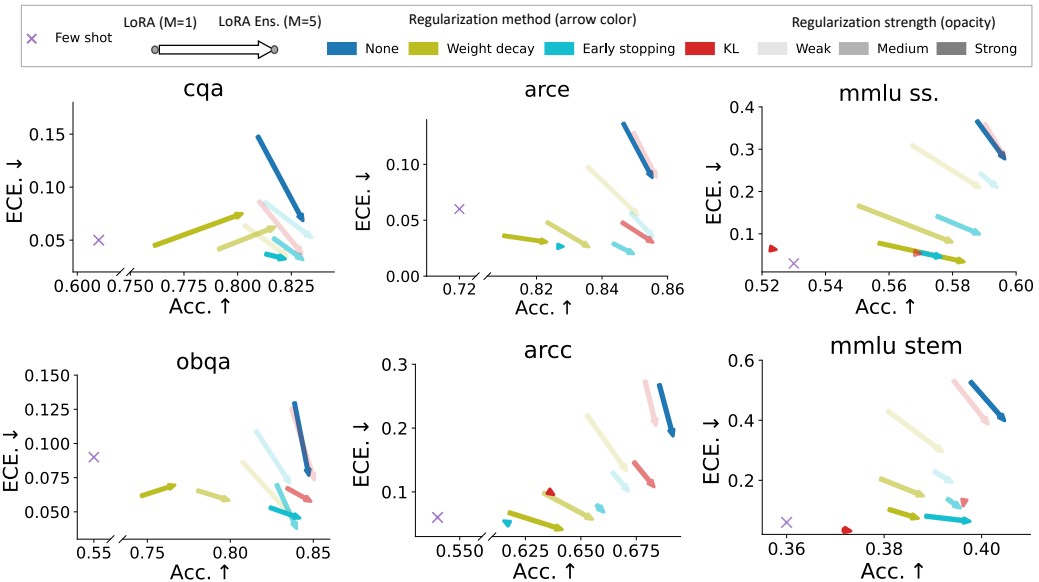

Figure 2: **LoRA ensembles improve both accuracy and calibration under different regularization techniques.** Arrows link the performance of a single LoRA model (arrow tail) to the corresponding ensemble with 5 LoRA components (arrowhead), where the x-axis denotes validation accuracy and the y-axis expected calibration error. Arrow colors indicate regularization methods and opacity reflects regularization strength. The majority of arrows are pointing toward the right bottom corner, suggesting that ensembling benefits both accuracy and calibration error measured by ECE.

weights, which requires them to store $M$ copies of the model, where $M$ is the number of ensemble components. This is impractical for modern LLMs, so instead they are forced to work with smaller models; in particular, they work with GPT-2 (Radford et al., 2019) with only 1.5 billion parameters. Hewitt et al. (2021) consider an ensemble of LLMs consisting of two components: a model trained with full fine-tuning, and a model trained with LoRA. In contrast, we consider ensembling with a large number of LoRA components (e.g. 20) to improve accuracy and calibration. There also exists an efficient ensemble method BatchEnsemble (Wen et al., 2020), where the ensemble components share a base model that is modified multiplicative by component-specific parameter-efficient rank-1 matrices, leading to reasonable storage demands comparable to our proposed LoRA ensembles. It has been applied on LLMs by Tran et al. (2022) but for pre-training rather than fine-tuning. While it may be possible to adapt BatchEnsemble to the fine-tuning setting, this has not, to our knowledge, yet been considered. Indeed, we believe that such an adaptation is a non-trivial exercise that may require careful consideration of e.g. how to initialize and train the multiplicative adapters to avoid "overpowering" the pre-trained weights.

**Calibration and uncertainty quantification of LLMs.** Pre-trained LLMs already show reasonably good calibration (OpenAI, 2023). Nonetheless, there are several recent papers that seek to further enhance *pre-trained* LLMs' calibration and uncertainty quantification ability in open-ended generation tasks. In particular, Lin et al. (2022); Kuhn et al. (2022) propose to use prompts to guide models to provide linguistically calibrated answers. Zhou et al. (2023) studies how LLMs express uncertainty in the natural language form. Our work is very different in that it focuses on mitigating very poor calibration that can emerge from fine-tuning.

## 3 BACKGROUND

### 3.1 FINE-TUNING LARGE LANGUAGE MODELS

Fine-tuning assumes access to a pre-trained LLM, denoted by by $\mathbf{W}^*$, usually an auto-regressive model based on the transformer architecture. In the tasks we consider, the fine-tuning data consists of prompts, $\mathbf{X} = \{\mathbf{x}_n\}_{n=1}^N$, and answers $\mathbf{y} = \{y_n\}_{n=1}^N$, where the prompt can, e. g., describe a multiple-

choice QA problem (Fig. 1b), and the answer can be in the label set $\mathcal{T} = \{ \text{"}a\text{"}, \text{"}b\text{"}, \text{"}c\text{"}, \text{"}d\text{"} \}$, which is a subset of all tokens $\mathcal{V}$ the LLM can generate. Given this data, fine-tuning entails initializing the parameters at $\mathbf{W} = \mathbf{W}^*$ and minimizing the loss $-\log p(\mathbf{y} \mid \mathbf{X}; \mathbf{W})$. In this paper, we consider tasks, where the label set $\mathcal{T} \subset \mathcal{V}$ of possible answers consists of single tokens.[1] Typically, there will be tokens $v \notin \mathcal{T}$ with nonzero probability under the LLM. To study calibration accuracy and predictive uncertainty of LLM fine-tuning, we introduce the normalized task distribution

$$p_{\mathcal{T}}(y \mid x_n; \mathbf{W}) = \begin{cases} p(y \mid x_n; \mathbf{W})/Z_{\mathbf{W}} & \text{if } \mathbf{y} \in \mathcal{T} \\ 0 & \text{otherwise,} \end{cases} \quad \text{where} \quad Z_{\mathbf{W}} = \sum_{\mathbf{y} \in \mathcal{T}} p(y \mid x_n; \mathbf{W}). \quad (1)$$

The normalized task distribution allows us to study the quality of predictions beyond accuracy.

## 3.2 Deep ensembles

A popular tool for improving the predictive uncertainty of deep learning methods are ensembles. Deep ensembles (Lakshminarayanan et al., 2017) offer a practical alternative for the fully Bayesian treatment of Bayesian neural networks (Neal, 2012) or Monte-Carlo Dropout (Gal & Ghahramani, 2016). They simply average the predictions of $M$ networks which have been trained separately using different random initializations,

$$p_{\text{ens}}(y \mid \mathbf{x}_n) = \frac{1}{M} \sum_{m=1}^{M} p(y \mid \mathbf{x}_n; \mathbf{W}_m). \quad (2)$$

## 3.3 Efficient Fine Tuning low-rank adapters (LoRA)

LoRA (Hu et al., 2021) is a parameter-efficient fine-tuning technique. Instead of fine-tuning all the model weights, it learns additive correction terms, called *adapters*, whose low-rank structure greatly reduces the number of trainable parameters. Each adapter $\Delta W = \alpha B A$ consists of trainable *low-rank* matrices $B \in \mathbb{R}^{d \times r}, A \in \mathbb{R}^{r \times k}$ of rank $r$ and a constant scaling factor $\alpha \in \mathbb{R}^+$, which is usually fixed. During fine-tuning, we fix $\mathbf{W}^*$, and only optimize $\Delta W$:

$$\mathcal{L}(\Delta W) = \min_{\Delta W} \sum_{n=1}^{N} -\log p(y_n \mid \mathbf{x}_n; \mathbf{W}^* + \Delta W). \quad (3)$$

Critically, $\Delta W$ represents far fewer parameters, so is much easier to fine-tune in constrained compute environments. As suggested by (Hu et al., 2021), it is common to initialize $A$ randomly and $B$ as zero, in that way, we can have $\mathbf{W}^* + \Delta W = \mathbf{W}^*$ at the beginning of the optimization, i.e. the fine-tuned model starts from the pre-trained model.

## 3.4 Regularization

**Output space regularization via KL regularization.** In LLM fine-tuning, it is common to include a KL regularization (Schulman et al., 2017; Bai et al., 2022; Ouyang et al., 2022; Korbak et al., 2022; He et al., 2022) to make the output distribution of the fine-tuned model close to that of the pre-trained model. In our setting, we consider the following KL regularization objective

$$\beta D_{KL}(p(\mathbf{y} \mid \mathbf{X}; W, \Delta W) \,\|\, p(\mathbf{y} \mid \mathbf{X}; W)), \quad (4)$$

which is added to Eq. (3) during optimization where $\beta$ controls the strength of the regularization.

**Implicit regularization via early stopping.** Another commonly adopted regularization method is early stopping. Early stopping halts the optimization when certain criteria are met such that the model is not over-optimized. The fewer epochs used, the stronger the regularization is.

## 4 Method

We propose LoRA ensembles, an ensemble of LLMs where each of the ensemble components is fine-tuned with LoRA. Remember that LoRA learns only low-rank additive correction terms, called

---

[1]Our method, LoRA ensembles also applies to open-ended generation tasks.

*adapters*, with a very small number of parameters. This makes LoRA a practical choice for LLM ensembles. Each ensemble component will use the same pre-trained weights $\mathbf{W}^*$, but will have its own adapter term $\Delta W_m$. This introduces strong parameter sharing between the component-specific weights $\mathbf{W}_m = \mathbf{W}^* + \Delta W_m$, and the adapter initialization becomes a source of diversity. After fine-tuning, LoRA facilitates efficient storage and retrieval, empowering us to swiftly execute ensembles at the prediction stage. Crucially, at test time, the large base model $\mathbf{W}^*$ is loaded only once, while the low-rank adapters can be loaded and unloaded with negligible overhead.

Importantly, LoRA ensembles can be applied on top of modified fine-tuning protocols, that include regularization. For example, these regularization methods might keep all the fine-tuned models in the ensemble close to the pre-trained model, which may additionally improve calibration over just the improvement from ensembling. In extensive experiments (Sec. 5.2), we find that ensembling can offer additional improvements in accuracy and calibration over those offered by regularization alone. Finally, we consider an additional form of regularization that we were not able to find explicitly discussed in prior work. In particular, we consider penalizing the LoRA $B$ matrix by including a very large weight decay term in AdamW (Loshchilov & Hutter, 2019). At the $t_{th}$ time step, the AdamW update for $B$ is

$$B_t \leftarrow B_{t-1} - \gamma(g_{t-1} - \lambda B_{t-1}). \tag{5}$$

Here, $\gamma$ is the step size, $g_{t-1}$ is the normalized gradient acquired from standard Adam (Kingma & Ba, 2014) with no regularization and $\lambda$ adjusts the strength of regularization. Although weight decay is a very standard regularization technique, we find that the usual setup with a $\lambda$ of $1e{-}2$ (the default setting from PyTorch's AdamW, denoted as "None" in our paper) barely helps resolve the overconfidence issue. Instead, we adopt an extreme large value of $\lambda$ ranging from $1e2$ to $1e3$. In addition, we adopt weight decay only on the $B$ matrix of $\Delta W$, which we notice shows the best performance (Appendix. B).

## 5 EMPIRICAL STUDY OF LORA ENSEMBLES

In this section, we evaluate LoRA ensembles on a collection of datasets to show its benefits in various QA settings (described in Sec. 5.1), and find that it leads to better predictive and calibration accuracy than baseline approaches (Sec. 5.2). In addition, we study the effect of regularization in LoRA ensembles, where we find that LoRA ensembles is complementary to many regularization techniques. Finally, we conduct ablation studies, to better understand the effect of our modeling choices on ensemble diversity and LoRA ensembles's performance (Sec. 5.3).

### 5.1 EXPERIMENTAL SET-UP

**Multiple-Choice Question Answering Datasets.** For our experiments, we choose six popular multiple-choice QA datasets for evaluation: CommonsenseQA (cqa, Talmor et al., 2019), Open-Book (obqa, Mihaylov et al., 2018), social sciences (mmlu ss.) and STEM (mmlu stem) subset from MMLU (Hendrycks et al., 2021), ARC-easy (arce) and ARC-challenge (arcc) from AI2 Reasoning Challenge (Clark et al., 2018). Questions in cqa have 5 options while the others all have 4 options. We provide details for the training and validation set of each task in Table 2 in the appendix, we provide example questions for each task in Appendix C.

**Evaluation metrics.** For all 6 tasks, we first measure the accuracy (Acc.) on the validation set. However, problems such as bad calibration or lack of uncertainty quantification can not be reflected through accuracy. Therefore we incorporate negative log-likelihood (NLL.), which measures the model uncertainty on held-out validation datasets, and expected calibration error (ECE., Guo et al., 2017) which assesses the alignment between predicted probabilities and actual empirical accuracy. Since safe deployment in real-world applications requires models to behave predictably when the data comes from another domain, we also study OOD performance. In particular, we test models fine-tuned on cqa on test samples from mmlu as OOD and we test models fine-tuned on a subset of mmlu with test samples from other mmlu subcategories. We then compute the accuracy, NLL., ECE., and additionally, the OOD detection performance measured by AUROC on the OOD test samples using negative maximum softmax probability (Hendrycks & Gimpel, 2016) as the score.

**Implementation Details of LoRA Ensembles.** We build LoRA ensembles by fine-tuning LLaMA-13b (Touvron et al., 2023) which has 13 billion parameters, where we use Transformers (Wolf et al.,

2020) and Huggingface PEFT (Mangrulkar et al., 2022) for model and LoRA implementations. In most experiments, we build ensembles with $M = 5$ components, though our ablation study also contains larger ensembles. As in Hu et al. (2021), we apply the adapter $\Delta W = \alpha BA$ only on the query and value matrices of the self-attention modules of LLaMA-13b and we fix $\alpha = 32$. With rank $r = 8$, each ensemble component has 6 million trainable parameters. The adapter matrices $B$ are initialized to be zero, while the entries of $A$ are randomly initialized using Kaiming Uniform (He et al., 2015). We use AdamW for all experiments and run optimizations for 20 epochs with a fixed step size of $5e-5$ . We use a batch size of 32 for cqa, 16 for obqa, arcc, and arce, and 8 for mmlu ss. and stem. Half-precision is used for all the forward and backward passes after which we convert the output logits to single precision when computing metrics. For all datasets, we experiment with four variations of LoRA ensembles: Default AdamW configuration with $\gamma = 0.01$, denoted as **None** in the figures; **KL** regularization from Eq. (4) with $\beta \in \{0.01, 0.05, 0.1\}$; **Early stopping** after $\{1, 2, 3\}$ epochs, and very large **weight decay** on $B$ from Eq. (5) with $\gamma \in \{1e2, 5e2, 1e3\}$.

We consider the following approaches as baselines

- **LoRA (M=1)** For all variations of LoRA ensembles, we report the averaged performance of the *single* ensemble members. We represent the results with solid lines in trace figures, in contrary to dashed lines for the ensembled versions.
- **Few shot** For each question in the validation set, we append $(\mathbf{X}, \mathbf{y})$ pairs from the training set in front of the prompts as "demonstration" to perform few shot learning (Brown et al., 2020). In our experiments, we randomly draw 3 pairs of $(\mathbf{X}, \mathbf{y})$ without replacement and evaluate the performance through the average of 10 random draws. We perform few shot experiments only on the pre-trained model without any fine-tuning.
- **Last-layer ensembles** Last-layer fine-tuning, also known as linear probing (Du et al., 2021; Wu et al., 2020), refers to freezing the model but only fine-tuning the last linear layer. We fine-tuned the rows in the linear head that correspond to the token for the options. multiple times starting from the pre-trained weights under different random seeds to construct an ensemble.
- **Monte Carlo (MC) dropout.** When dropout is employed at training time, we can use MC dropout (Gal & Ghahramani, 2016) to perform ensembling: Instead of training multiple LoRA adapters, we can train a single one and keep Dropout on at test time and perform multiple forward passes with nodes randomly shut down. MC dropout has previously been adopted in masked language models for incorporating uncertainty estimation (Sankararaman et al., 2022; Vazhentsev et al., 2022). We combine dropout with *standard* LoRA fine-tuning by adding dropout on the input of the LoRA adapter following the implementation of Mangrulkar et al. (2022).

## 5.2 RESULTS

In Fig. 2, we present the validation accuracy and ECE. of different LoRA ensembles fine-tuning methods after 20 epochs. Critically, ensembling usually gives considerable improvements in accuracy and calibration compared with the single-component versions regardless of the regularization method or strength. LoRA ensembles also shows significantly improved accuracy compared with few shot learning (purple "x"), confirming the value of fine-tuning. Fig. 2 also shows that regularization usually improves calibration, as measured by ECE. However, the effect of regularization on accuracy is more inconsistent: stronger regularization often reduces accuracy (e.g. stronger regularization in arce reduces accuracy) though sometimes increases accuracy (mmlu stem; early stopping).

Next, we look at the behavior of (regularized) LoRA ensembles across training (Fig. 3). This reinforces the results from Fig. 2. In particular, LoRA ensembles consistently improves both calibration and accuracy, whether applied with or without regularization (here, weight decay). However, weight decay has conflicting effects: it seems to usually reduce accuracy while improving calibration. Interestingly, the NLL metric seems to become very large for several of the datasets (e.g. mmlu ss. and stem). This is likely because the NLL heavily penalizes overconfidence: assigning very, very low probability to the right answer. Interestingly, ensembling on its own was not sufficient to prevent this dramatic increase in NLL, while weight decay was sufficient to prevent the dramatic increase (though weight decay in combination with ensembling consistently gave the best NLL).

We present the performance of MC dropout in Fig. 4. We find that MC dropout shows a marginal improvement over the performance of a single model while LoRA ensembles gives dramatically larger improvements in terms of both accuracy and calibration. The performance of last-layer ensembles is

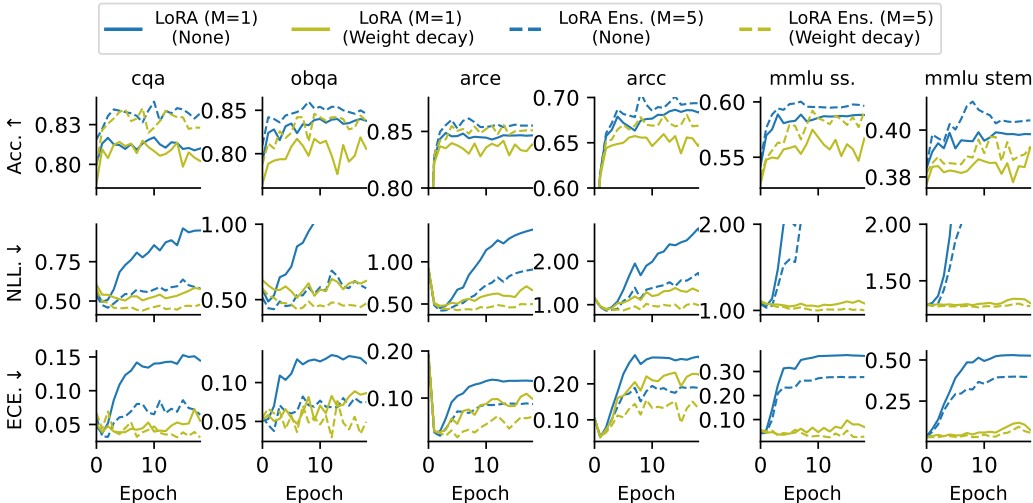

Figure 3: **LoRA ensembles improves accuracy while regularization prevents NLL from blowing up.** For all ensemble results we use $M = 5$ components. We use $\lambda = 1e3$ for mmlu subsets and $\lambda = 1e2$ for others for weight decay.

Table 1: Performance of last-layer Ensembles. Last-layer Ensembles underfit the data, showing accuracy significantly worse than LoRA ensembles.

|  | Acc. ↑ | | NLL. ↓ | | ECE. ↓ | |
|---|---|---|---|---|---|---|
|  | LoRA Ens. | Last-layer Ens. | LoRA Ens. | Last-layer Ens. | LoRA Ens. | Last-layer Ens. |
| cqa | 0.83 | 0.52 | 0.58 | 1.25 | 0.06 | 0.06 |
| obqa | 0.85 | 0.48 | 0.58 | 1.27 | 0.07 | 0.12 |
| arce | 0.86 | 0.72 | 0.92 | 0.79 | 0.09 | 0.06 |
| arcc | 0.69 | 0.48 | 1.46 | 1.30 | 0.19 | 0.15 |
| mmlu ss. | 0.60 | 0.46 | 2.72 | 1.47 | 0.28 | 0.19 |
| mmlu stem | 0.41 | 0.32 | 4.04 | 1.72 | 0.40 | 0.25 |

presented in Table 1, which also shows worse accuracy than LoRA ensembles. Fine-tuning only the linear head might not be expressive enough for downstream tasks adaptation. Lastly, we find that ensembling is also helpful in the OOD setting (Fig. 5). While it fails to resolve catastrophic forgetting in cqa v.s. mmlu, it shows improvements both in terms of NLL. and ECE., providing more reliable predictions on unseen domains.

## 5.3 ADDITIONAL ABLATIONS

In this section, we perform extra ablations to gain a better understanding of the effect of the number of ensemble components and the randomness sources on the performance LoRA ensembles.

**Number of ensemble components.** To start with, we study the effects of ensemble component number $M$, we show the results in Fig. 6 where we experiment with LoRA ensembles under different numbers of ensemble components. We collect 20 ensemble components in total and we report the average results of 5 random draws from them for each $M$. We notice that increasing $M$ improves all metrics however the marginal benefit of increasing components diminishes as $M$ becomes larger.

**Ensemble diversity under different regularization strengths.** In Fig. 2, high strength of KL regularization and early stopping could cause the performance gain of ensembling to vanish. This is not surprising in that KL regularization directly forces all ensemble components to make predictions similar to the pre-trained model while early stopping prevents different ensemble models from further moving away from the pre-trained model. Weight decay suffers the least from this problem, we suspect that this is caused by the complicated relationship between the weight space and the output space as well as we are performing optimization long enough for ensemble members to diverge.

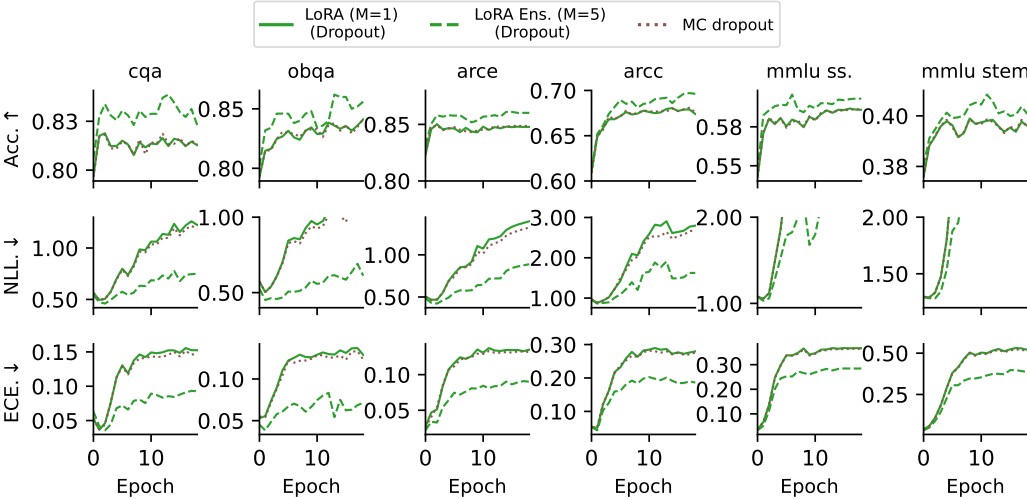

Figure 4: **Ensemble of LoRA significantly outperforms MC dropout under the same number of ensemble members.** When employing dropout during the fine-tuning, an alternative ensemble strategy becomes available: Keeping dropout on at test time to implement Monte Carlo (MC) dropout. However, MC dropout offers only marginal performance gains compared to a standalone model, outperformed by ensembles of independently trained LoRA models when both methods employ the same number of ensemble members (chosen as 5 in our experiments).

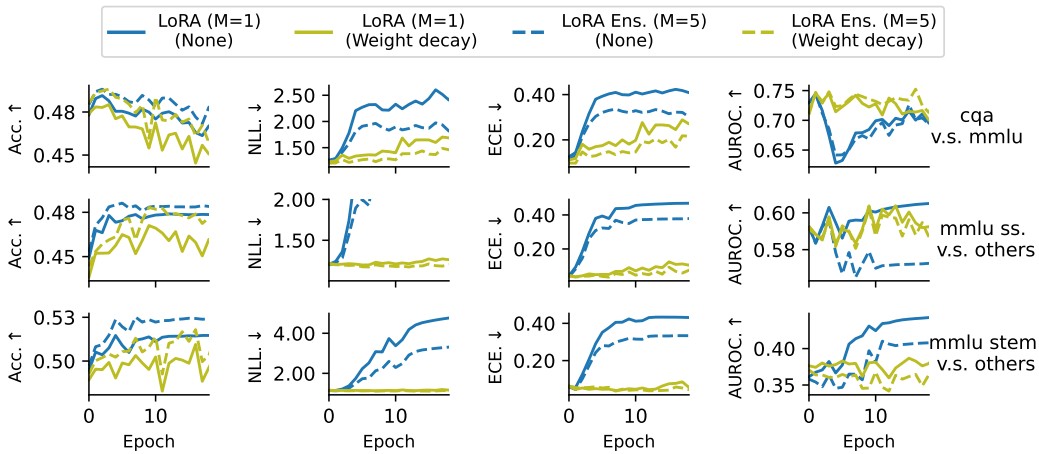

Figure 5: **Ensembles offer benefits for accuracy and calibration over regularized and unregularized fine-tuning approaches in OOD settings.** Note that all methods show AUROC around or lower than $0.5$ on the second and third row, we suspect the models would fail to *detect* OOD samples if they can *generalize* to them, as the accuracy increases throughout fine-tuning.

**Ensemble diversity under different sources of randomness.** Next, we study the source of randomness in LoRA ensembles. As discussed in previous sections, the diversity of LoRA ensembles comes from two sources: The random initialization and the randomness from dataset shuffling (i.e. SGD noise). It is often observed that random initialization contributes mostly to the diversity of ensemble (Fort et al., 2019). However, it is unclear whether this is the case for LoRA ensembles. To investigate, we conduct experiments on cqa under three settings: Dataset shuffling with fixed initialization, random initialization with fixed dataset shuffling, and no randomness (both fixed). For each setting, we conduct experiments with 5 independent trials and the results are presented in Fig. 7. We observe that LoRA ensembles can work with either source of randomness alone, while randomness from dataset shuffling (i.e. SGD noise) contributes more to the ensemble performance.

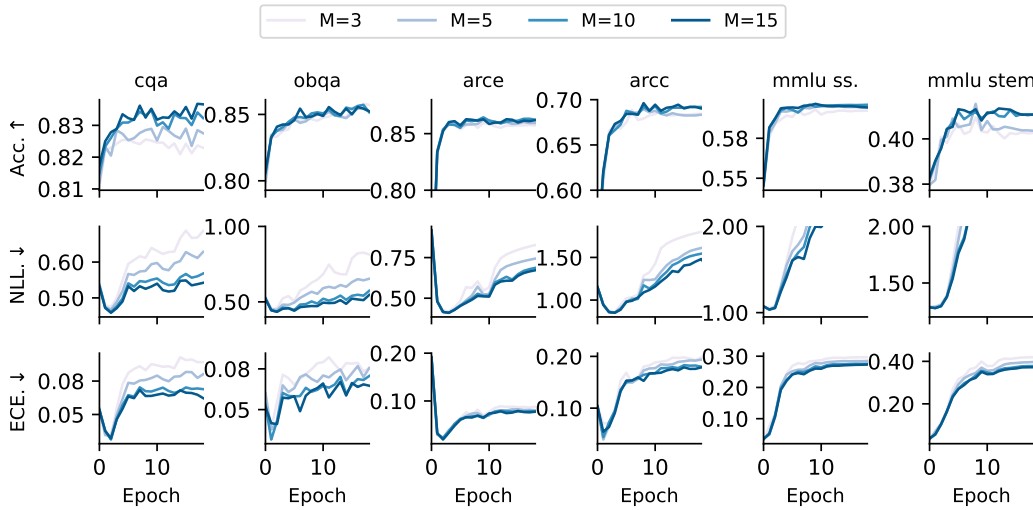

Figure 6: **Increasing the number of ensembles improves accuracy and calibration.** With LoRA ensembles, we can efficiently ensemble with a large number of components, however, the performance gains from increasing the number of components become less substantial with larger $M$.

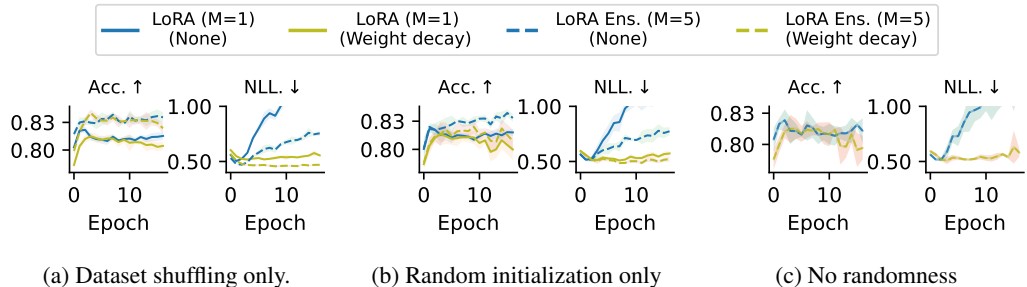

(a) Dataset shuffling only.    (b) Random initialization only    (c) No randomness

Figure 7: **Randomness from initialization and dataset shuffling both contribute to the diversity of LoRA ensembles.** The diversity of ensembles can be reflected by the gap between the dashed lines and solid lines. When regularization is used, SGD noise from dataset shuffling alone is more beneficial than random initialization alone.

However, it is hard to decompose exactly the contribution of each source of randomness, and in practice, one should incorporate both random initialization and dataset shuffling for better diversity.

## 6 DISCUSSION

In this paper, we develop a new method for LLM fine-tuning: LoRA ensembles. Our empirical results on 6 datasets demonstrate that our proposed method improves both the accuracy and calibration of fine-tuning a single model. In addition, we propose to combine regularization techniques together with ensembling for better calibration. Broadly, LLMs have demonstrated their power in a variety of scenarios, but their safety issues have started to draw more and more attention (Wei et al., 2023; Jones & Steinhardt, 2022; Perez et al., 2022): Real-world applications require not only the LLM to be accurate but also reliable. Our method provides a key ingredient towards addressing these concerns, in that it helps LLMs to make not only precise but also calibrated predictions.

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

# A    ADDITIONAL IMPLEMENTATION DETAILS

We provide a summary of the datasets information in Table. 2.

Table 2: Summary of task setting. For mmlu, we combine the development and validation set as the training set and use the origin test split as the validation set, for the rest datasets, we use the default training set and test splits (validation split for cqa) as the validation set.

| Task | Size of training set | Size of validation set | Number of options |
|------|---------------------|------------------------|-------------------|
| cqa | 8741 | 1221 | 5 |
| obqa | 4957 | 500 | 4 |
| arce | 2249 | 2375 | 4 |
| arcc | 1119 | 1172 | 4 |
| mmlu ss. | 397 | 3077 | 4 |
| mmlu stem | 411 | 3018 | 4 |

# B    ABLATION STUDY ON WEIGHT DECAY REGULARIZATION

Note that LoRA is defined as $\Delta W = BA$. Therefore we have three ways options for applying weight decay regularization: Regularizing only $A$, regularizing only $B$, or regularizing both. We experiment with these three options on cqa, mmlu ss. and mmlu stem. where we choose $\gamma = 1e2$ for cqa and $\gamma = 1e3$ for mmlu. We present the performance of these three strategies on validation set in Fig. 8, where we find that regularizing only $A$ results in high NLL., regularizing both matrices causes significant drops in accuracy, while regularizing only $B$ provides satisfying results in both accuracy and NLL.

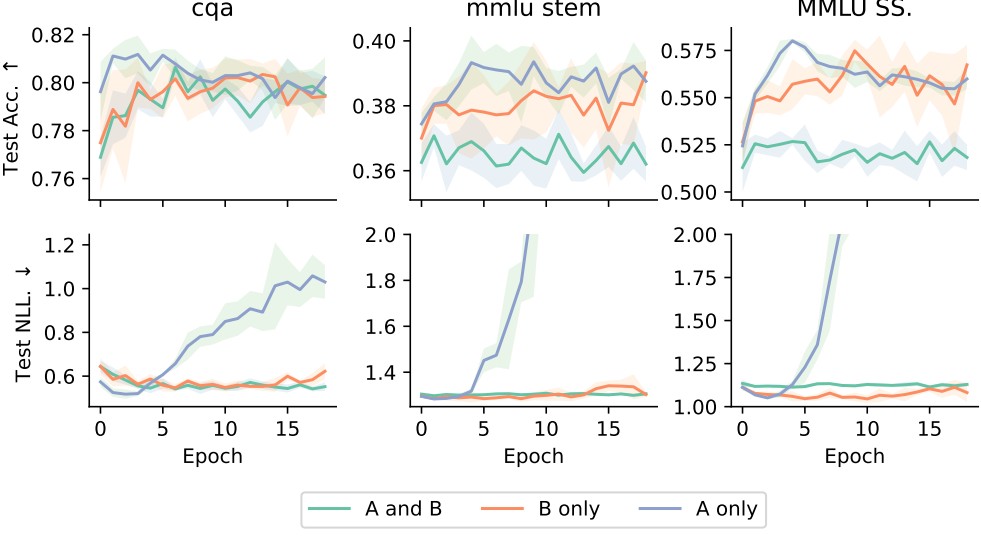

Figure 8: Very large weight decay on A and B underfits the data while regularizing only LoRA A does not resolve overconfidence. Regularizing only LoRA B shows the best performance.

## C  SAMPLE QUESTIONS

In this section, we provide sample questions from the datasets we experiment with.

### C.1  COMMONSENSE QA (CQA)

```
Q: The sanctions against the school were a punishing blow,
and they seemed to what the efforts the school had made to change?
Answer Choices:
(a) ignore
(b) enforce
(c) authoritarian
(d) yell at
(e) avoid
A: (a).
```

```
Q: Sammy wanted to go to where the people were.
Where might he go?
Answer Choices:
(a) race track
(b) populated areas
(c) the desert
(d) apartment
(e) roadblock
A: (b).
```

```
Q: To locate a choker not located in a jewelry box or
boutique where would you go?
Answer Choices:
(a) jewelry store
(b) neck
(c) jewlery box
(d) jewelry box
(e) boutique
A: (a).
```

### C.2  OPENBOOK QA (OBQA)

```
Q: The sun is responsible for
Answer Choices:
(a) puppies learning new tricks
(b) children growing up and getting old
(c) flowers wilting in a vase
(d) plants sprouting, blooming and wilting
A: (d)
Q: When standing miles away from Mount Rushmore
Answer Choices:
(a) the mountains seem very close
(b) the mountains are boring
(c) the mountains look the same as from up close
(d) the mountains seem smaller than in photographs
A: (d)
Q: When food is reduced in the stomach
Answer Choices:
(a) the mind needs time to digest
(b) take a second to digest what I said
(c) nutrients are being deconstructed
(d) reader's digest is a body of works
```

A: (c)

## C.3 ARC-EASY (ARCE)

Q: Which factor will most likely cause a person to develop
a fever?
Answer Choices:
(a) a leg muscle relaxing after exercise
(b) a bacterial population in the bloodstream
(c) several viral particles on the skin
(d) carbohydrates being digested in the stomach
A: (b)
Q: Lichens are symbiotic organisms made of
green algae and fungi. What do the green algae supply
to the fungi in this symbiotic relationship?
Answer Choices:
(a) carbon dioxide
(b) food
(c) protection
(d) water
A: (b)
Q: When a switch is used in an electrical circuit, the switch can
Answer Choices:
(a) cause the charge to build.
(b) increase and decrease the voltage.
(c) cause the current to change direction.
(d) stop and start the flow of current.
A: (d)

## C.4 ARC-CHALLENGE (ARCC)

Q: George wants to warm his hands quickly by rubbing them.
Which skin surface will produce the most heat?
Answer Choices:
(a) dry palms
(b) wet palms
(c) palms covered with oil
(d) palms covered with lotion
A: (a)
Q: Which of the following statements best explains
why magnets usually stick to a refrigerator door?
Answer Choices:
(a) The refrigerator door is smooth.
(b) The refrigerator door contains iron.
(c) The refrigerator door is a good conductor.
(d) The refrigerator door has electric wires in it.
A: (b)
Q: A fold observed in layers of sedimentary rock most
likely resulted from the
Answer Choices:
(a) cooling of flowing magma.
(b) converging of crustal plates.
(c) deposition of river sediments.
(d) solution of carbonate minerals.
A: (b)

## C.5 MMLU SOCIAL SCIENCES (MMLU SS.)

Q: What should a public relations media practitioner do if

she does not know the answer to a reporter's question?
Answer Choices:
(a) Give the reporter other information she is certain is correct.
(b) Say that the information is 'off the record' and will
    be disseminated later.
(c) Say 'I don't know' and promise to provide the
    information later.
(d) Say 'no comment,' rather than appear uninformed.
A: (c).

Q: In issues management, what is the most proactive approach to
addressing negative or misleading information posted online about
your organization?
Answer Choices:
(a) Buy domain names that could be used by opposition groups.
(b) Post anonymous comments on blogs to combat this information.
(c) Prepare a news release that discredits the inaccurate
    information.
(d) Make policy changes to address complaints highlighted on
    these sites.
A: (d).

Q: Which of these statements is true of the Vatican in 2010 at
the time of the accusations of child abuse cover-ups?
Answer Choices:
(a) There was a coordinated media response.
(b) Consistent messages were communicated.
(c) Criticisms were taken as attacks on the Catholic Church.
(d) The credibility of the Vatican was upheld.
A: (c).

## C.6   MMLU STEM (MMLU STEM)

Q: Let V be the set of all real polynomials p(x). Let
transformations T, S be defined on V by T:p(x) -> xp(x)
and S:p(x) -> p'(x) = d/dx p(x), and interpret (ST)(p(x))
as S(T(p(x))). Which of the following is true?
Answer Choices:
(a) ST = 0
(b) ST = T
(c) ST = TS
(d) ST - TS is the identity map of V onto itself.
A: (d).

Q: A tank initially contains a salt solution of 3 grams
of salt dissolved in 100 liters of water. A salt solution
containing 0.02 grams of salt per liter of water is sprayed into
the tank at a rate of 4 liters per minute.  The sprayed solution
is continually mixed with the salt solution in the tank, and the
mixture flows out of the tank at a rate of 4 liters per minute.
If the mixing is instantaneous, how many grams of salt are in
the tank after 100 minutes have elapsed?
Answer Choices:
(a) 2
(b) 2 - e^-2
(c) 2 + e^-2
(d) 2 + e^-4
A: (d).

Q: Let A be a real 2x2 matrix. Which of the following statements
must be true?
I. All of the entries of A^2 are nonnegative.
II. The determinant of A^2 is nonnegative.
III. If A has two distinct eigenvalues, then A^2 has two
distinct eigenvalues.
Answer Choices:
(a) I only
(b) II only
(c) III only
(d) II and III only
A: (b).

