# OpenReview forum: "LoRA ensembles for large language model fine-tuning"
_ICLR.cc/2024/Conference — Submitted to ICLR 2024_

### Official Review · Reviewer_QbwE · 2023-11-01

**Soundness:** 3 good
**Presentation:** 3 good
**Contribution:** 2 fair
**Rating:** 5
**Confidence:** 4

**Summary:**

The paper introduces LoRA ensembling to improve LLMs' performance and uncertainty calibration. The core idea is to train multiple LoRA adapters and ensemble them to get more accurate and calibrated predictions, as typically shown in deep ensembling literature. Experiments results verify that LoRA ensembling improves over baseline approaches.

**Strengths:**

- The method is built on top of well-known results that model ensembles can lead to more accurate and calibrated predictions.
- LoRA ensemble alleviates the need to finetune and update the entire model which is computationally prohibitive.
- Experiment results show that LoRA ensemble does lead to more accurate results as well as reduced calibration error.

**Weaknesses:**

- It is rather straightforward to consider LoRA finetuning for ensembling. The technical novelty of the proposed method is a bit limited.
- There are several relevant and stronger baselines not considered in the experiments, including calibration for in-context learning [1], and self-consistency [2], both of which shows decent improvements on prediction accuracy.
- The current set of experiments considered is limited to multiple choice questions (predicting only a single token). While the method is indeed compatible with generative tasks, there are no such tasks considered in the experiments. The results would be more convincing with generative tasks since LLMs are commonly used for complex tasks.
- LoRA ensemble requires finetuning datasets. However, LLMs are commonly used in zero or few-shot ways in many scenarios. How does the method perform if only a few finetuning data is available?

[1] Calibrate Before Use: Improving Few-Shot Performance of Language Models. Zhao et al. 2021.

[2] Self-Consistency Improves Chain of Thought Reasoning in Language Models. Wang et al. 2022.

**Questions:**

- The presentation of experiment results are scattered. It would be easier to compare different methods if they are all listed in the same table/figure. Currently, comparisons to different baselines are in separate tables/figures, making it a bit hard to read.
- In Figure 3, 4, 5, and 6, why ECE increases as number of epoch increases? In this case, isn't the method making calibration worse by LoRA finetuning?
- In many cases, finetuning on specific dataset can compromise the model's performance on other tasks. Does LoRA ensemble suffer the same problem?

---

> ### Author Response · Authors · 2023-11-16
> **Author response**
>
> ## Novelty of the method
>
>  Please refer to the general response.
>
> ## Comparision with self-consistency baseline
>
> Self-Consistency [1] shows improvement upon chain-of-thought, however, as a few-shot-learning-based approach, it is outperformed by LoRA ensemble, which utilizes model fine-tuning, we provide a comparison of accuracy based on the numbers from Table. 3 in [1].
> | Method                           | Commonsense QA | ARC-C | ARC-E |
> |----------------------------------|----------------|-------|-------|
> | LoRA Ensemble with Llama-13B     | 83             | 69    | 86    |
> | Self-Consistency with LaMDA-137B | 63.1           | 59.8  | 79.3  |
>
> where LoRA ensembles outperform Self-Consistency with a backbone model ten times smaller.
>
> ## Multiple choice QA tasks
>
> The main reason we choose to stick to multiple-choice QA is that the metric for measuring calibration and uncertainty quantification is well-defined. On open-ended generative tasks, how to measure output's calibration and uncertainty remains an open question. For example, [3] shows that there are multiple ways to quantify the uncertainty in open-ended QA problems while each metric could have different behaviors. We consider combining these advanced metrics with LoRA ensemble as a future research problem.
>
>
> ## Scenarios with limited fine-tuning data
>
> When there is only a limited amount of fine-tuning data available, the fine-tuned model would demonstrate severe overconfidence, as we can see from the experiment results from MMLU social sciences and STEM experiments, where there are only 397 and 411 training samples available: The single model (LoRA M=1) shows high ECE and NLL while LoRA ensembling significantly resolves the overconfidence. Our results show that the performance can be further improved by combining regularization with LoRA ensemble.
>
> ## Presentation of the results
>
> Thanks for the suggestion, we plan to add a table that summarizes the results from all different methods and baselines in the appendix for easier comparison.
>
> ## Increasement calibration error
> Note that the training objective for LoRA fine-tuning is cross-entropy loss, therefore the fine-tuned model would tend to keep reducing the training loss by making the predictive distribution sharper, this is fine on the training set, as the model can reach %100 accuracy, however on the test set, the model would suffer from *increasing* calibration error (ECE) and negative log-likelihood (NLL) as the model is making overconfident and wrong predictions. Our results show that this can be alleviated by using LoRA ensemble, while weight decay regularization can further prevent the ECE from growing up.
>
>
> ## Performance degradation on other tasks
> The top row in Figure. 5 shows the performance on MMLU from the model fine-tuned on commonsense QA. We can see that, while both the single LoRA and LoRA ensembles show decreased accuracy as fine-tuning proceeds, LoRA ensembles still show improvement of performance upon the single fine-tuned model, we believe this implies that LoRA ensembles suffer less from performance degradation.
>
> ---------------
> [1] Calibrate Before Use: Improving Few-Shot Performance of Language Models. Zhao et al. 2021.
>
> [2] Self-Consistency Improves Chain of Thought Reasoning in Language Models. Wang et al. 2022.
>
> [3] Kuhn, Lorenz, Yarin Gal, and Sebastian Farquhar. "Semantic uncertainty: Linguistic invariances for uncertainty estimation in natural language generation." arXiv preprint arXiv:2302.09664 (2023).

---

> > ### Comment · Reviewer_QbwE · 2023-11-20
> > **Thank you for the response**
> >
> > Thank you to the authors for the response.
> >
> > For comparison to other baselines like self-consistency, a fair comparison would be self-consistency or calibration on a LoRA
> > fine-tuned model.
> >
> > Overall, I still have concern on the novelty of the method and the need of finetuning data for the method to work. I would remain my score.

---

### Official Review · Reviewer_o2gF · 2023-11-02

**Soundness:** 2 fair
**Presentation:** 2 fair
**Contribution:** 2 fair
**Rating:** 5
**Confidence:** 4

**Summary:**

This conference paper discusses issues of Large Language Models (LLMs) in overconfidence and uncertainty in their predictions. To mitigate these issues, the paper proposes a new approach called LoRA ensembles, which leverages low-rank adapters and random initialization to create diverse model components. This method addresses the limitations of traditional ensemble approaches, such as excessive storage requirements and a lack of diversity in fine-tuned LLMs. The authors demonstrate the effectiveness of LoRA ensembles in improving accuracy and calibration for various reasoning tasks compared to alternative fine-tuning methods and introduce the concept of regularized LoRA ensembles to further enhance performance and address potential correlations between components. This research enables the scaling of ensemble methods to LLMs with billions of parameters, offering a promising solution for enhancing the reliability of LLM predictions in safety-critical applications like medical diagnosis, finance, and decision-making processes.

**Strengths:**

(1) The paper is well-written and easy to understand.

(2) This paper analyses potentials of LoRA ensemble with various techniques, such as regulizers, Dropout, weight decay.

(3) The ablation study of LoRA with randomness is interesting.

**Weaknesses:**

(1) One major concern is the idea is very naive and straightforward. The performance improvement of deep ensemble is already well-known  to the community, and it is in no way surprising that we can combine LoRA with ensemble to improve the performance, uncertainty, etc.

(2) Another concern is that no computational costs is reported in this paper. I understand the inference costs of LoRA ensemble is much lower than traditional finetuning ensemble, but it is good to demonstrate this.

(3) The title in the paper doesn't exactly match the title in the openreview system.

(4) Figure 7 is not easy to understand. It is better to put all variants in one figure or use tables to compare the performance of various combination of methods.

**Questions:**

Please see the above weaknesses.

---

> ### Author Response · Authors · 2023-11-16
> **Author response**
>
> ## Novelty of the method
>
> Please refer to the general response.
>
> ## Computational cost
>
> We provide a summarization of loading time and memory cost below under Llama-13b and LoRA of rank 8. Note that the memory requirements for full-model ensembling are beyond the capacity of an 80GB A100 GPU, therefore at test time, one may need to load and unload the models for ensembling, causing extra overhead.
>
> |              | Full model ensemble (M=5) | LoRA ensemble (M=5) |
> |--------------|---------------------------|---------------------|
> | Loading time | ~30s                      | ~6.5s               |
> | Memory usage | ~129GB                     | ~26GB               |
>
> ## Discrepancy in title
> We apologize for the mistake, we will fix the title in later revisions.

---

### Official Review · Reviewer_zRFn · 2023-11-03

**Soundness:** 3 good
**Presentation:** 2 fair
**Contribution:** 2 fair
**Rating:** 3
**Confidence:** 4

**Summary:**

The paper argued ensembling LLMs is computationally challenging due to the sheer size of such models. In light of this, the authors proposed a solution to create an ensemble of models with Low-Rank Adapters (LoRA), referred to as LoRA ensembles. These ensembles are much smaller in terms of parameters and can be efficiently constructed on top of underlying pre-trained models. The results demonstrate that LoRA ensembles, when applied independently or in combination with other regularization techniques, offer improved predictive accuracy and achieve better uncertainty quantification.

**Strengths:**

- The introduction of LoRA for ensembling is a unique approach, particularly for large models like LLMs. This could be a  useful exploration of how to ensemble such massive models.
- LoRA ensembles, whether used independently or in conjunction with other techniques, demonstrate enhancements in both prediction accuracy and the quantification of uncertainty.
- The observation that regularization may benefit calibration over just the improvement from ensembling can hold practical value.

**Weaknesses:**

- The paper lacks a comprehensive survey of existing ensemble methods, and it does not adequately discuss or compare with related works such as [1,2,3,4,6] in the literature.
- The focus of the paper is only on prediction ensembles, which neglects the important weight ensemble methods [1,3,4,6]. The paper argues that maintaining an ensemble of, for instance, 5 LLMs in memory can be challenging in certain scenarios. However, it's worth noting that weight ensembles require the maintenance of just one model. Recent papers adopt online ensemble methods [3] that continuously average weight parameters.
- The concept of ensembling adapters in LLMs has been previously explored in [5], yet this prior work is neither discussed nor compared in the paper.
- The method's evaluation is restricted to small datasets, and its scalability remains unverified. Furthermore, the absence of actual ensemble baselines is notable. For example, [1,3] employ ensemble techniques while training the model only once, which is highly relevant to the task addressed in this paper.

A minor issue is the presence of a discrepancy between the title displayed on OpenReview and the actual title in the paper.

[1] Deep Ensembling with No Overhead for either Training or Testing: The All-Round Blessings of Dynamic Sparsity. ICLR 2021

[2] Training Independent Subnetworks for Robust Prediction. ICLR 2020

[3] SWAD: Domain Generalization by Seeking Flat Minima. NeurIPS 2021

[4] DNA: Domain generalization with diversified neural averaging. ICML 2022

[5] AdapterSoup: Weight Averaging to Improve Generalization of Pretrained Language Models. EACL 2023

[6] Averaging Weights Leads to Wider Optima and Better Generalization. UAI 2018

**Questions:**

Please refer to the weaknesses mentioned above.

---

> ### Author Response · Authors · 2023-11-16
> **Author response**
>
> ## Weight averaging v.s. (output space) ensembling.
>
> Thanks for pointing out the related literature on weight averaging. We will add discussion to weight space averaging methods in later revisions.  It is important to clarify, though, that we use the term "ensemble" to represent output space ensembling following classic literature ([2]), we are unaware of the potential confusion with weight space averaging during writing, which we will further clarify in later revisions.
>
> We opted not to include weight averaging methods as a baseline for several reasons. Firstly, in studies focusing on ensembling to enhance neural networks' calibration and uncertainty quantification, such as [4, 5, 6], weight averaging methods like SWA [7] are rarely used as a baseline. The reason is that weight space averaging aims at solving problems fundamentally different from output space ensembling: Weight space averaging methods target acquiring models with **better generalization**, which is usually measured by accuracy. In contrast, output space ensembling methods, including Bayesian neural networks, MC dropout, and deep ensembles, aim to address issues of overconfidence and poor calibration, evaluated using metrics like NLL and ECE. Notably, weight-averaging methods are **not** recognized for resolving calibration and overconfidence issues, with calibration error and out-of-distribution detection performance seldom serving as evaluation metrics in weight-averaging studies.
>
> Moreover, an examination of Tables 3, 4, and 5 in the appendix of [1] reveals that SWA's calibration error is consistently surpassed by output space ensembling methods. To further investigate the impact of weight averaging on LLM fine-tuning, we applied SWA to LoRA by averaging weights across five iterations. Our results, detailed below, indicate that while SWA does enhance accuracy, it fails to improve NLL and ECE. This suggests that SWA is not effective in addressing the overconfidence and calibration problems central to our study."
>
>
> | Method              | Accuracy | Negative log-likelihood (NLL) | Expected calibration error (ECE)|
> |---------------------|----------|-------------------------|----------------------------|
> | LoRA (M=1)          | 0.808    | 0.948                   | 0.153                      |
> | LoRA (M=1, SWA)     | 0.813    | 1.18                    | 0.156                      |
> | LoRA Ensemble (M=5) | **0.832**    | **0.581**                   | **0.064**                      |

---

> ### Author Response · Authors · 2023-11-16
> **Author response cont.**
>
> ## Scalability of the method.
>
> It is unclear to us what type of "scalability" the reviewer refers to. The biggest dataset we used is commonsense QA, which has 9.8K training samples. With LoRA ensemble, we are also able to perform large-scale ensembling experiments using 15 ensemble components (Fig. 6), which is not possible for full model fine-tuning. We also did not see any obstacles in applying LoRA ensembles to larger datasets or models without trouble. Essentially, if LoRA can be applied for fine-tuning, LoRA ensemble can be applied to boost the performance. We are happy to incorporate more experiments if the reviewer could elaborate more on scalability.
>
> ##  Baseline on ensemble method.
>
> In the "Ensembling of Neural Networks" and "Ensembling in LLMs" paragraphs in the related work section, we discussed works on ensembling methods. However, these methods only consider training ensembles from scratches and it is unclear how one can apply these methods for LLM fine-tuning: Adapting these methods to LLM fine-tuning is beyond the scope of our paper. We are also not aware of any works that perform (output space) ensembling for LLM fine-tuning, we are happy to include these methods as baselines if the reviewer could kindly point out references to these works.
>
> ## Relationship with AdapterSoup
>
> We apologize for missing the reference. However, our method differs from AdapterSoup in the following aspects: 1. AdapterSoup focuses on improving out-of-domain generalization while LoRA ensembles focus on improving overconfidence and calibration; 2. AdapterSoup trains multiple adapters on different domains starting from the same random seed while LoRA ensemble trains multiple adapters on a single task using different random seeds;  3. AdapterSoup uses the adapter proposed by [8] while LoRA ensembles use low-rank adapters and it is unclear (from the paper) whether AdapterSoup can alleviate overconfidence and bad calibration.
>
>  -----------
>
> [1] Maddox, Wesley J., et al. "A simple baseline for bayesian uncertainty in deep learning." Advances in neural information processing systems 32 (2019).
>
> [2] Sollich, Peter, and Anders Krogh. "Learning with ensembles: How overfitting can be useful." Advances in neural information processing systems 8 (1995).
>
> [3] AdapterSoup: Weight Averaging to Improve Generalization of Pretrained Language Models. EACL 2023
>
> [4] Wen, Yeming, Dustin Tran, and Jimmy Ba. "Batchensemble: an alternative approach to efficient ensemble and lifelong learning." arXiv preprint arXiv:2002.06715 (2020).
>
> [5] Izmailov, Pavel, et al. "What are Bayesian neural network posteriors really like?." International conference on machine learning. PMLR, 2021.
>
> [6] Florian Wenzel, Jasper Snoek, Dustin Tran, and Rodolphe Jenatton. Hyperparameter ensembles for robustness and uncertainty quantification. Advances in Neural Information Processing Systems, 33:6514–6527, 2020b.
>
> [7] Averaging Weights Leads to Wider Optima and Better Generalization. UAI 2018
>
> [8] Bapna, Ankur, Naveen Arivazhagan, and Orhan Firat. "Simple, scalable adaptation for neural machine translation." arXiv preprint arXiv:1909.08478 (2019).

---

### Author Response · Authors · 2023-11-16
**Genearl response**

General response:

We would like to thank the reviewers for the detailed review!

Our submission proposes LoRA ensembles, which adopt Deep Ensemble to large language model fine-tuning settings using low-rank adapters. LoRA ensembles can efficiently and effectively resolve the overconfidence and bad calibration of fine-tuned LLM, as the reviewers agree:
- zRFn: The introduction of LoRA for ensembling is a unique approach, particularly for large models like LLMs. This could be a useful exploration of how to ensemble such massive models.
- QbwE: Experiment results show that LoRA ensemble does lead to more accurate results as well as reduced calibration error.


Here, we provide a clarification on the *novelty* of our method: We consider simplicity as one of our method's biggest advantages. We believe LoRA ensemble can serve as a robust and simple way to improve the calibration and uncertainty quantification for fine-tuned LLM. As our experiment results illustrate, without any effort tuning hyper-parameters, out-of-box LoRA ensembles can consistently improve the expected calibration error and accuracy across all tasks. In addition, despite that the idea of performing ensembling for LLM fine-tuning using low-rank adapters seems obvious, we are not aware of any existing works that propose similar ideas: Previous works either consider ensemble over full models [1, 2], which is less scalable and efficient than our method, or consider using Monte Carlo dropout [3], which is outperformed by LoRA ensemble as we illustrate in Fig. 4 of the manuscript. In addition, existing works do not consider combining regularization techniques and ensembling, which we find to be crucial for the fine-tuned model to reach the calibration of few-shot learning. Reviewer zRFn points out that there is a recent work proposing to average the weights of multiple adaptation layers acquired on different tasks to make fine-tuned language models generalizable to different domains, our work differs from their work in that we focus on improving the calibration and uncertainty quantification of fine-tuned LLM rather than generalization ability.

-------------

[1] Adam Gleave and Geoffrey Irving. Uncertainty estimation for language reward models. arXiv preprint arXiv:2203.07472, 2022.

[2] Meiqi Sun, Wilson Yan, Pieter Abbeel, and Igor Mordatch. Quantifying uncertainty in foundation models via ensembles. In NeurIPS 2022 Workshop on Robustness in Sequence Modeling, 2022.

[3] Karthik Abinav Sankararaman, Sinong Wang, and Han Fang. Bayesformer: Transformer with uncertainty estimation. arXiv preprint arXiv:2206.00826, 2022.

---

### Meta-Review · Area_Chair_Uobo · 2023-12-04

**Metareview:**

This paper investigates ensembles of Low-Rank Adapters (LoRA) for a single LLM to achieve the benefits of ensembling without an intractable computational overhead. The paper is well written and the results demonstrate the benefits of LoRA ensembles; however, there are concerns about novelty and limited experiments. The idea presented in this paper is a rather straightforward combination of two techniques (LoRA and ensembling), has many similarities to prior work (especially the AdapterSoup paper), and the results are unsurprising (ensembling is beneficial). The authors claim in their rebuttal that the simplicity of their method is a strength, and I am inclined to agree. However, simple and straightforward methods require a larger “burden of proof,” especially when the results are largely expected. The author’s analysis was limited to relatively simple datasets, did not compare against many relevant baselines, and did not verify scalability with larger experiments. Therefore, this paper is not ready for publication at ICLR in its current state.

**Justification For Why Not Higher Score:**

All of the reviewers had similar concerns (lack of experiments, straightforward method with expected results), and no one recommended publication. After reading the reviews, author discussion, and skimming the paper myself, I am inclined to agree.

**Justification For Why Not Lower Score:**

N/A

---

### Decision · Program_Chairs · 2024-01-16

Reject